# Exploration of Correlations between COVID-19 Vaccination Choice and Public Mental Health Using Google Trend Search

**DOI:** 10.3390/vaccines10122173

**Published:** 2022-12-17

**Authors:** Shao-Cheng Wang, Yuan-Chuan Chen

**Affiliations:** 1Department of Psychiatric, Taoyuan General Hospital, Ministry of Health and Welfare, Taoyuan 33004, Taiwan; 2Department of Mental Health, Johns Hopkins Bloomberg School of Public Health, Baltimore, MD 21205, USA; 3Department of Nursing and Department of Medical Technology, Jenteh Junior College of Medicine, Nursing and Management, Miaoli County 35664, Taiwan; 4Program in Comparative Biochemistry, University of California, Berkeley, CA 94720, USA

**Keywords:** COVID-19, Pfizer/BNT vaccine, Moderna vaccine, AstraZeneca vaccine, google trend, public mental health

## Abstract

Since the global COVID-19 pandemic has great impact on human health and life style, the vaccination is the most effective method for disease control and prevention. However, not all people are willing to be vaccinated because some critical factors affect vaccination aspiration and vaccine choice of the public population. Among these factors, public mental health belongs to a political issue. In this study, Google Trend Search was used to explore the correlation between COVID-19 vaccination choice and public mental health during the period from August/2020 to December/2021. The results suggested that the main public concerns of COVID-19-related mental illnesses are positively correlated with the new cases amount but are negatively correlated with total cases and vaccinated cases amount. Moreover, the results support that the public population took more interest in the Pfizer/BNT COVID vaccine and Moderna COVID vaccine than the AstraZeneca COVID vaccine. Our study shows that investigations of the public mental health should be set up and conducted widely. A complete vaccination program combined with a policy for the improvement of public mental health are very effective for the control and prevention of COVID-19.

## 1. Introduction

The disease caused by the new coronavirus SARS-CoV-2 was designated by the World Health Organization (WHO) on 11 February 2020 as coronavirus infectious disease in 2019 (COVID-19), and its pandemic has seriously threatened human health worldwide [1,2]. SARS-CoV-2 may cause mild, moderate, or severe symptoms in humans and even lead to death in some severe cases. Additionally, COVID-19 is raising global human anxiety and fear, consequently causing public concerns of mental illnesses. The evidence shows that infected patients experience neuropsychiatric complications including anxiety disorders, mood disorders, and suicidal ideation, and even convalescent COVID-19 patients are at high risk of depression [3,4]. Furthermore, COVID-19 patients are at high risk of neurological illnesses such as headache, sleep disorders, encephalopathy, and loss of taste and smell [5]. Therefore, COVID-19 is an infectious disease with latent neuropsychiatric consequences.

Since the COVID-19 outbreaks in the world, vaccination has become the most effective method for prevention and provides a clear path to bring the pandemic to an end. Although the advantages are shown to be significantly greater than the disadvantages for vaccination based on the scientific benefit/risk evaluation, some people are still hesitant in choosing what kind of vaccines or even choosing not to be vaccinated. Generally, the reasons why some people have hesitancy for vaccination or even refuse to be vaccinated are as follows: (1) Concerns of vaccine effectiveness and safety: The evaluation results of vaccine efficacy and safety are variable in different studies, depending on regions, countries, races, genders, ages, and individuals. For example, the same COVID-19 vaccine demonstrates significant difference in efficacy and side effects in different clinical trials. The severity of side effects is usually especially related to personal genetics. (2) Fears of possible critical adverse reactions: Based on some special clinical studies and case reports, COVID-19 vaccines may result in rare but severe adverse reactions. Some critical adverse reactions may be serious and even lethal. (3) Emergency of virus variants: The evolution of SARS-CoV-2 is so fast that virus variants emerge rapidly and frequently. The condition for pathogens to evade the protection of vaccines is possible. Many people are still infected with viruses by breakthrough infection after vaccination, so some people consider vaccination as useless.

Moreover, the political view of matters such as public mental health may also be a critical factor affecting the choice of vaccination or not. Consequently, it is an utterly important issue to determine how people choose vaccines, why some people refuse to be vaccinated, and the role of political views in making these decisions because these choices may significantly interfere with the promotion of vaccination and the control of the COVID-19 pandemic. The goal of our study is to explore the correlation between public mental health and the decision for COVID-19 vaccination and the consequences of these decisions. 

## 2. Materials and Methods

Currently, several COVID-19 vaccines, such as Pfizer/BNT, Moderna, AstraZeneca, Johnson & Johnson, BBIBP-CorV, CoronaVac, and Novavax, have been well-developed and approved for emergency-use authority (EUA). Among them, the Pfizer/BNT, Moderna, and AstraZeneca vaccines are three major vaccines approved for clinical application globally during 2020 and 2021 (Table 1). Many scientific publications for the evaluation of these three major COVID-19 vaccines had already been published [6,7,8,9,10,11,12]. These three vaccines significantly dominate the media, Internet, and our real life to an unprecedented degree.

The academic community, the public, and the general population all need more COVID-19 vaccine information. Google Trend has already proved to be a tool for documentation and scientific review associated with COVID-19. For example, Springer et al. revealed that Google Trend is a successful tool for monitoring population concerns and information needs during the COVID-19 pandemic [13]. Moreover, using Google Trend, they demonstrated that the interest of the population focused on the therapeutic options rather than theories related to a COVID-19 animal origin [14]. Therefore, Google Trend has potentially been used as a tool to evaluate population concerns and information needs for COVID-19 vaccines [15,16].

Regarding the mental illness related to COVID-19, we used the search terms “COVID anxiety”, “COVID depression”, “COVID-19 delta”, “COVID-19 omicron”, “total cases”, “new cases”, and “total vaccination” to document a dominant information for the population during the prevalence of two different risky variants of COVID-19: Delta and Omicron. The Pearson correlations were calculated for these seven terms from 23 August 2020 to 23 November 2021. Further search terms of “Pfizer/BNT COVID vaccine”, “Moderna COVID vaccine”, “AstraZeneca COVID vaccine”, “total cases”, “new cases”, and “total vaccination” can document the dominant information needed for individuals to protect themselves by vaccine. The Pearson correlations were calculated for these six terms. Finally, we used the search terms “Anxiety”, “Depression”, “COVID-19 vaccine”, “total cases”, “new cases”, and “total vaccination” to document the dominant information regarding mental illness available for the population to search in conjunction with COVID-19 vaccine information. The Pearson correlations were calculated for these six terms from August 2020 to December 2021.

## 3. Results

The results of the search terms “COVID anxiety”, “COVID depression”, “COVID-19 delta”, “COVID-19 omicron”, “total cases”, “new cases”, and “total vaccination” are shown in Figure 1. The search for “COVID-19 delta” reached its peak on 1 August 2021 and then decreased sharply. The search for “COVID-19 omicron” reached its peak on 28 November 2021. The search for “COVID-19 anxiety” and “COVID-19 depression” gradually decreased from 23 August 2020 to 23 November 2021.

The assessment of the Pearson correlation coefficient shown in Table 2 reveals a highly positive correlation between total cases and total vaccination cases (r = 0.9708, *p* < 0.00), a negative correlation between the number of total vaccinations cases and the search terms “COVID anxiety” (r = −0.5425, *p* < 0.00) and “COVID depression” (r = −0.7374, *p* < 0.00), and a negative correlation between the number of total cases and the search terms “COVID anxiety” (r = −0.5772, *p* < 0.00) and “COVID depression” (r = −0.7536, *p* < 0.00) as time passed. In the same setting, the number of new COVID-19 cases shows significantly positive correlation results between “COVID anxiety” (r = 0.6054, *p* < 0.00) and “COVID depression” (r = 0.6327, *p* < 0.00).

From Figure 1 and Table 2, we obtain the information as follows: (1) The total vaccination would be increased if total cases increased. (2) COVID anxiety and COVID depression would be decreased if the total vaccination increased. (3) COVID anxiety and COVID depression would be increased if new COVID-19 cases increased. (4) COVID anxiety and COVID depression would decrease as time passed even if the total cases increase. These results suggest that as the total number of COVID-19 cases increase, the vaccination cases are also increasing worldwide. Furthermore, the searches for “COVID-19 anxiety” and “COVID-19 depression” increased as well with increasing new cases but decreased as time passed.

Further search terms including “Pfizer/BNT COVID vaccine”, “Moderna COVID vaccine”, “AstraZeneca COVID vaccine”, “total cases”, “new cases”, and “total vaccination” can document the dominant information individuals searched for to protect themselves by vaccine. As shown in Figure 2, a worldwide search revealed the significant peaks in the searches for these three terms.

The “Pfizer/BNT vaccine” search culminated around 9 November 2020. Later, the “Pfizer/BNT vaccine” search decreased to the bottom rankings around 30 November 2020 and then rose to its second significant peak on 14 December 2020, with a third significant peak on 18 January 2021, a fourth significant peak around 15 April 2021 (most), and a fifth significant peak around on 23 August 2021. The “Moderna vaccine” search demonstrated a first significant peak around 16 November 2020. Though the “Moderna vaccine” search was down to the bottom of the rankings on 30 November 2020 and gradually rose during 16–30 November 2020, it gradually increased and maintained about six peaks from 14 December 2020 to 15 February 2021. In 2021, two significant peaks appeared around 31 March and 9 August, respectively. The “AstraZeneca vaccine” search revealed three small peaks around 23 November 2020, 28 December 2020, and 1 February 2021 and reached its most significant peak on 14 March 2021, respectively. From then on to 5 December 2021, its search gradually decreased (Figure 2).

Overall, the “Pfizer/BNT vaccine” search was the most popular, the “Moderna vaccine” search was the second, and the “AstraZeneca vaccine” was the least. Furthermore, search terms may reflect an increased concern and information need regarding these three vaccines worldwide. 

The assessment of the Pearson correlation coefficient revealed a correlation among the number of total COVID-19 cases and the search terms “Moderna COVID vaccine” (r =−0.6818, *p* < 0.00), “Pfizer/BNT COVID vaccine” (r = −0.6490, *p* < 0.00), and “AstraZeneca COVID vaccine” (r = −0.5406, *p* < 0.00). In the same setting, the search for the number of new COVID-19 cases showed similar medium correlation results. This suggests that as the number of COVID-19 cases increases, the worldwide interest or concern for these COVID-19 vaccines also increases. For the Pearson correlation coefficient, a negative correlation between the number of individuals with vaccinations and the number of new cases (r = −0.4165, *p* < 0.00) was found, suggesting that as the number of individuals with vaccinations increase, new cases decrease. Furthermore, a negative correlation between the number of total vaccinations and the search term “Moderna COVID vaccine” (r =−0.7423, *p* < 0.00) was found; a more negative correlation for “AstraZeneca COVID vaccine” (r = −0.6849, *p* < 0.00) was also found. These results suggest that more and more people searched “Moderna COVID vaccine” on Google instead of “AstraZeneca COVID vaccine” as the number of total vaccinations was increasing (Table 3).

The results of search terms “Anxiety”, “Depression”, “COVID-19 vaccine”, “total cases”, “new cases”, and “total vaccinations” are shown in Figure 3. The searches for “COVID-19 vaccine” decreased gradually after 1 August 2021 as well as the searches for “Anxiety” and “Depression”.

The assessment of the Pearson correlation coefficient is shown in Table 4 and revealed a negative correlation among search terms on “COVID-19 vaccine” and “Anxiety” (r = 0.7312, *p* < 0.00) and “Depression” (r = 0.4975, *p* < 0.00). This suggests that those people who searched more information about “COVID-19 vaccine” are probably at high risk of anxiety or depression.

## 4. Conclusions

COVID-19 is not only an infectious disease but a disease with potential neuropsychiatric consequences. Furthermore, the social behaviors related to COVID-19, including lockdown, mask wearing, and Internet surfing, interact with public mental health [2]. In this study, we used the scores of Google Trend searches to represent the public concerns, particularly about public mental health. Additionally, using Google Trend search, we analyzed the correlations among the public concerns of mental health, the public concerns about the COVID-19 vaccine, and the number of COVID-19 cases and vaccinations. To the best of our knowledge, this is the first study to explore the correlation between COVID-19 cases; the public concerns of COVID-19 vaccines, including the “Pfizer/BNT COVID vaccine”, “Moderna COVID vaccine”, and “AstraZeneca COVID vaccine”; and the public concerns of COVID-19-related mental illnesses, including “COVID Anxiety”, “COVID Depression”, “Anxiety”, and “Depression” using Google Trend search. Therefore, Google Trend search is a possible approach to explore the public mental health.

Our results suggest that the main public concerns of COVID-19-related mental illnesses are positively correlated with the number of new cases but negatively correlated with the number of total cases and vaccinated cases. This reveals that public anxiety and depression increased with the most recent emergence of highly risky SARS-CoV-2 variants and decreased with the greater availability of vaccinations and/or as more and more infected people gained immunity against SARS-CoV-2. Eventually, the population may feel less anxious and depressed after we have more available effective medical treatments and effective strategies to reduce transmission of COVID-19.

Furthermore, by the data from August 2020 to December 2021, the results support that the main interest of the population is in the “Pfizer/BNT COVID vaccine” and “Moderna COVID vaccine” compared with the “AstraZeneca COVID vaccine”, which may be due to the higher protection rate of the former two vaccines compared to the latter vaccine (95% vs. 70%) (Table 1). This is commensurate with the current condition of vaccine prevalence in the world: mRNA vaccines such as the “Pfizer/BNT COVID vaccine” and “Moderna COVID vaccine” are more popular than adenovirus-carried vaccines such as the “AstraZeneca COVID vaccine”. In addition, the scores of Google Trend for “Pfizer/BNT COVID vaccine”, “Moderna COVID vaccine”, and “AstraZeneca COVID vaccine” searches are all highly positive correlated, suggesting that they are all widely known to be available and effective for COVID-19 prevention. The public’s mental health concerns are highly related to the scores of Google Trend searches for “COVID-19 vaccine”, suggesting the public anxiety and fear prior to or after vaccination. We suggested that individuals will have anxiety, fear, and/or other mood symptoms before they receive COVID-19 vaccination, and these results are consistent with the concept of “vaccine hesitancy” [17]. COVID-19 patients who are experiencing neuropsychiatric complications, including direct and indirect effects on the central nervous system, have been recognized [2,18]. Our findings suggest that public mental health, including concerns of anxiety and/or depression, is affected by COVID-19, as is the individual’s mental health both before and after vaccination. The current effective interventions for COVID-19 include vaccinations for prophylaxis and antiviral drugs for therapeutics [19].

It cannot be denied that people’s choice to be vaccinated or not and choice of vaccine are dependent on information from public concerns and Google searches. However, some information in the media may be indefinite, confusing, or even lead to misunderstandings for the public. For example, some recently published papers demonstrated that the spike protein of SARS-CoV-2 alone may cause acute respiratory distress syndrome, lung endothelial barrier dysfunction, as well as changes in other organs. These studies have focused on developing animal models for studying the virus, but they became the subject of speculation about the dangers of RNA vaccines in social media and thus could also influence the choice to receive vaccination or not. Thus, political issues, including public mental health, region, economics (poverty or wealthy), inequality, and education, are also likely to affect vaccination rates. Our study results support that public mental health programs for the population should be established and implemented extensively to improve the public’s psychological wellbeing. Vaccination guidelines only have been insufficient to prevent disease transmission. A complete vaccination program combined with the policy of public mental health improvement are an actually effective strategy against COVID-19.

## Figures and Tables

**Figure 1 vaccines-10-02173-f001:**
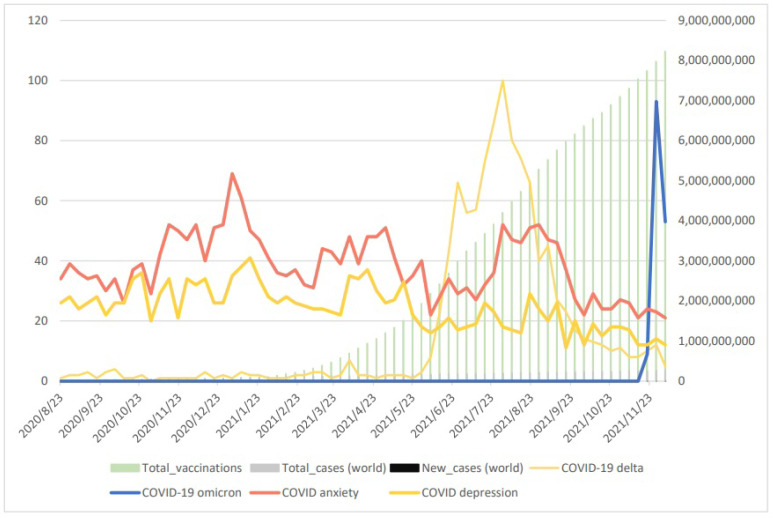
Display of Google Trend data for two major COVID-19 variants, i.e., Delta and Omicron, and COVID-19 anxiety and depression as well as for total and new COVID-19 cases. Normalized search data were obtained from Google Trend (100—high interest; 0—no or insufficient interest data) for the period from 23 August 2020 to 5 December 2021. COVID-19 case data were from JHU (John Hopkins University) CSSE COVID-19 Data (accessed 5 December 2021).

**Figure 2 vaccines-10-02173-f002:**
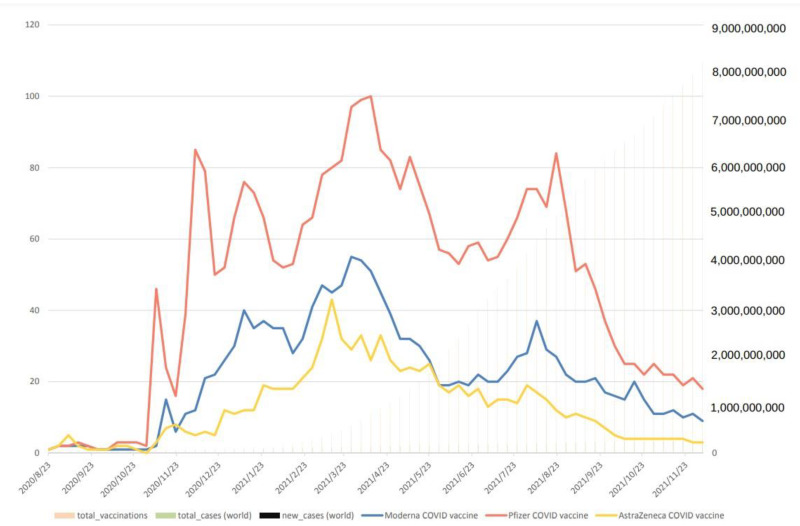
Display of Google Trend data for three major COVID-19 vaccines, including the Moderna vaccine, Pfizer/BNT vaccine, and AstraZeneca vaccine, as well as for total and new COVID-19 cases. Normalized search data were obtained from Google Trend (100—high interest; 0—no or insufficient interest data) for the period from 23 August 2020 to 5 December 2021. COVID-19 case data were from JHU (John Hopkins University) CSSE COVID-19 Data (accessed on 5 December 2020).

**Figure 3 vaccines-10-02173-f003:**
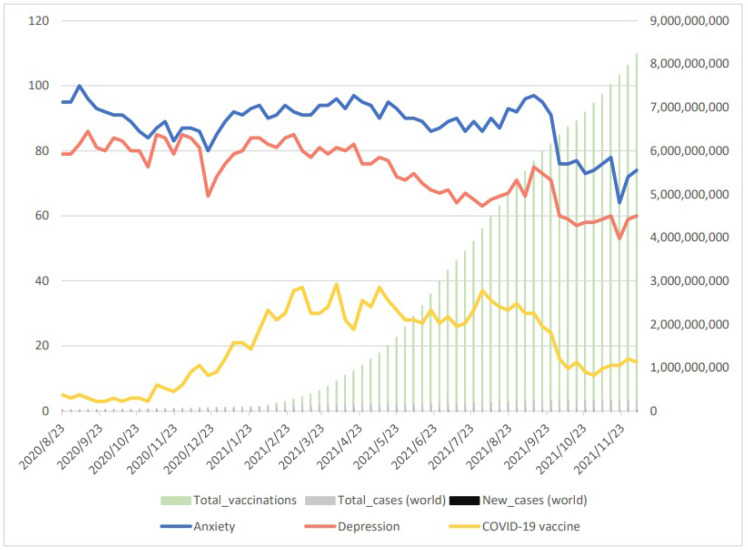
Display of Google Trend data for COVID-19 vaccines, anxiety, and depression as well as for total and new COVID-19 cases. Normalized search data were obtained from Google Trend (100—high interest; 0—no or insufficient interest data) for the period from 23 August 2020 to 5 December 2021. COVID-19 case data were from JHU (John Hopkins University) CSSE COVID-19 Data (accessed on 5 December 2020).

**Table 1 vaccines-10-02173-t001:** Comparison of Pfizer/BNT vaccines, Moderna vaccines, and AstraZeneca vaccines.

Vaccine Item	Pfizer/BNT Vaccine	Moderna Vaccine	AstraZeneca
Manufacturer	Pfizer Inc. and BioNTech Inc.	Moderna Inc. and NIH USA	AstraZeneca Inc. and Oxford University
Vaccine type	mRNA	mRNA	Engineered adenovirus
Dosage (interval)	2 doses (21 days)	2 doses (28 days)	2 doses (28 days)
Storage condition (period)	−70 °C (180 days); 2–8 °C (5 days)	−20 °C (30 days)	2–8 °C (180 days)
Alleged protection rate	95%	95%	Average 70%, Highest 90%
Reference	[6,7]	[8,9]	[10,11,12]

Abbreviation: National Institute of Health, United States of America (NIH USA); messenger RNA (mRNA); Incorporation (Inc.).

**Table 2 vaccines-10-02173-t002:** The Pearson correlations among total vaccination, total cases, new cases, COVID-19 Delta, COVID-19 Omicron, COVID anxiety, and COVID depression.

Vaccine Case	Total Vaccination	Total Cases	New Cases	COVID-19 Delta	COVID-19 Omicron	COVID Anxiety	COVID Depression
Total vaccination	1						
Total cases	0.9708 *	1					
New cases	−0.4165 *	−0.3836 *	1				
COVID-19 Delta	0.3356 *	0.3750 *	−0.1682	1			
COVID-19 Omicron	0.3728 *	0.3276 *	−0.0301	−0.0794	1		
COVID anxiety	−0.5425 *	−0.5772 *	0.6054 *	0.0451	−0.2871 *	1	
COVID depression	−0.7374 *	−0.7536 *	0.6327 *	−0.2997 *	−0.2755 *	0.6636 *	1

Note: * *p* < 0.05.

**Table 3 vaccines-10-02173-t003:** The Pearson correlations between, total vaccinations, total cases, new cases, Moderna, Pfizer/BNT, and AstraZeneca.

Vaccine Case	Total Vaccination	Total Cases	New Cases	Moderna	Pfizer/BNT	AstraZeneca
Total vaccination	1					
Total cases	0.9708 *	1				
New cases	−0.4165 *	−0.3836 *	1			
Moderna	−0.7423 *	−0.6818 *	0.5131 *	1		
Pfizer/BNT	−0.7408 *	−0.6490 *	0.6245 *	0.8535 *	1	
AstraZeneca	−0.6849 *	−0.5406 *	0.3597 *	0.8380 *	0.7871 *	1

Note: * *p* < 0.05.

**Table 4 vaccines-10-02173-t004:** The Pearson correlations between total vaccination, total cases, new cases, anxiety, depression, and COVID-19 vaccine.

Vaccine Case & Mental Health	Total Vaccination	Total Cases	New Cases	Anxiety	Depression	COVID-19 Vaccine
Total vaccination	1					
Total cases	0.9708 *	1				
New cases	−0.4165 *	−0.3836 *	1			
Anxiety	−0.6512 *	−0.5460 *	0.3775 *	1		
Depression	−0.8621 *	−0.8242 *	0.3808 *	0.7992 *	1	
COVID-19 vaccine	−0.3858 *	−0.2143	0.2295	0.7312 *	0.4975 *	1

Note: * *p* < 0.05.

## Data Availability

Database were from Google trend search and WHO COVID-19 database.

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
