# Peer review of "Exploration of Correlations between COVID-19 Vaccination Choice and Public Mental Health Using Google Trend Search"

_vaccines, 2022, doi:10.3390/vaccines10122173_

Round 1

Reviewer 1 Report

The article is very interesting and the methodology novel. The information you could potentially share will be relevant. The great work of the researchers should be recognized. However, it is necessary to modify some parts of the methodology so that the conclusions are solid and supported by results.

According to, the objective of the study is " improve the understanding of how and why public mental health are associated with COVID-19 vaccination”. Suggest answering the how and why, but only generate a hypothesis based on the analysis of correlations. Additionally, an additional analysis of the correlations is required that would not allow reaching these conclusions, perhaps the authors should use a pathway analysis or define the achievement of the objectives. This could is the major issue of this article.

Additionally, there are some fixes that suggest improvement.

1. Table 1, I suggest that it be moved after the title of the methodology, since it visually confuses the reader in relation to the end of the introduction.

2. In line 56, "be vaccinated, Generally, ", a period is missing.

3. The paragraph from line 56 to 69 is long and confusing. I consider it essential for understanding the problem.

4. Major correction. To include multivariate analysis is a must otherwise the study will be only be descriptive study.

I am not an expert in English edition, but I found the article difficult to read. Some editing service could be suggested.

Author Response

The article is very interesting and the methodology novel. The information you could potentially share will be relevant. The great work of the researchers should be recognized. However, it is necessary to modify some parts of the methodology so that the conclusions are solid and supported by results.

According to, the objective of the study is " improve the understanding of how and why public mental health are associated with COVID-19 vaccination”. Suggest answering the how and why, but only generate a hypothesis based on the analysis of correlations. Additionally, an additional analysis of the correlations is required that would not allow reaching these conclusions, perhaps the authors should use a pathway analysis or define the achievement of the objectives. This could be the major issue of this article.

 A: We have revised " improve the understanding of how and why public mental health are associated with COVID-19 vaccination and the consequences of these decisions.” into “explore the correlation between the public mental health and the decisions of COVID-19 vaccination and the consequences of these decisions. We hope this revision can be consistent with the title and the conclusion of this article.

Additionally, there are some fixes that suggest improvement.

  1. Table 1, I suggest that it be moved after the title of the methodology, since it visually confuses the reader in relation to the end of the introduction.

Ans: We have moved Table 1 and its related description to the first paragraph of Materials and Methods” section.

  1. In line 56, "be vaccinated, Generally, ", a period is missing.

Ans. We have added a period to the end of this sentence.

  1. The paragraph from line 56 to 69 is long and confusing. I consider it essential for understanding the problem.

Ans: We revised the these sentence a little bit. The aim for us to write the paragraph from line 56 to 69 is to provide the possible reasons for some people to have hesitancy for vaccination or even refuse to be vaccinated. By this paragraph, we introduced that the political view like public mental health may also be a critical factor affecting the choice of vaccination or not in next paragraph.

  1. Major correction. To include multivariate analysis is a must otherwise the study will be only be descriptive study.

 Ans: We greatly appreciate the reviewer’s comment and suggestion. Yes, this is only a descriptive study. It is complex job and will take us a lot of time to include multivariate analysis in this manuscript. Therefore, we will prepare a new study contain multivariate analysis for next published paper.

I am not an expert in English edition, but I found the article difficult to read. Some editing service could be suggested.

Ans: We have tried our best to check English language and asked a person whose native language is English to check the language for us. Moreover, the English service and editing will be done by our team in case of acceptance for publication.

Reviewer 2 Report

1) After reading the manuscript, I have recognized that the main result of the research is that the public mental health should be set up and conducted widely and that a complete vaccination program combining with the policy for the improvement of public mental health are really effective for the control and prevention of COVID-19.

2) the topic is actual.   3) The research has been considered the political issues related with a massive vaccination campaign, in my experience this is the first study which includes into consideration that part of public life.   4) So far, this is the first study of such a kind, I hope that this research shall open a new direction on the studies of a massive vaccination campaigns.   5)The conclusion consistent with the presented evidences.   6)All the references are appropriate.   7)  The tables and figures are OK, it is not necessary to improve it.

Author Response

1) After reading the manuscript, I have recognized that the main result of the research is that the public mental health should be set up and conducted widely and that a complete vaccination program combining with the policy for the improvement of public mental health are really effective for the control and prevention of COVID-19.

2) the topic is actual.   3) The research has been considered the political issues related with a massive vaccination campaign, in my experience this is the first study which includes into consideration that part of public life.   4) So far, this is the first study of such a kind, I hope that this research shall open a new direction on the studies of a massive vaccination campaigns.   5)The conclusion consistent with the presented evidences.   6)All the references are appropriate.   7)  The tables and figures are OK; it is not necessary to improve it.

Ans: We thanks for the reviewer’s positive comment. We will try our best to revise this manuscript to have a perfect paper for publication.

Reviewer 3 Report

The manuscript of Wang and Chen is well-written and is of high interest to the scientific community from public mental health and also to biologists studying the SARS-CoV-2 virus. Nevertheless, the authors missed one important search trend - the action of SARS-CoV-2 Spike protein. Some recently published studies demonstrated that coronavirus's spike protein alone may cause acute respiratory distress syndrome, lung endothelial barrier dysfunction, as well as changes in other organs. These studies have focused on developing animal modes for studying the virus, but they became the subject of speculation about the dangers of RNA vaccines in social media and could also influence the choice of the vaccine. This point should be discussed in the manuscript.

Author Response

The manuscript of Wang and Chen is well-written and is of high interest to the scientific community from public mental health and also to biologists studying the SARS-CoV-2 virus. Nevertheless, the authors missed one important search trend - the action of SARS-CoV-2 Spike protein. Some recently published studies demonstrated that coronavirus's spike protein alone may cause acute respiratory distress syndrome, lung endothelial barrier dysfunction, as well as changes in other organs. These studies have focused on developing animal models for studying the virus, but they became the subject of speculation about the dangers of RNA vaccines in social media and could also influence the choice of the vaccine. This point should be discussed in the manuscript.

Ans: We have added a few sentences to discuss this issue in the “Conclusion” section.

Round 2

Reviewer 1 Report

Thanks for attending to the suggestions.